# *Allium sativum* Extract Chemical Composition, Antioxidant Activity and Antifungal Effect against *Meyerozyma guilliermondii* and *Rhodotorula mucilaginosa* Causing Onychomycosis

**DOI:** 10.3390/molecules24213958

**Published:** 2019-10-31

**Authors:** Marcel Pârvu, Cătălin A. Moţ, Alina E. Pârvu, Cristina Mircea, Leander Stoeber, Oana Roşca-Casian, Adrian B. Ţigu

**Affiliations:** 1Department of Biology, Faculty of Biology and Geology, Babeș-Bolyai University, 42 Republicii Street, 400015 Cluj-Napoca, Romania; mircea.i.cristina@gmail.com (C.M.);; 2Department of Chemistry, Faculty of Chemistry and Chemical Engineering, Babeș-Bolyai University, 11 Arany Janos Street, 400028 Cluj-Napoca, Romania; augustinmot@chem.ubbcluj.ro; 3Department of Pathophysiology, Faculty of Medicine, Iuliu Haţieganu University of Medicine and Pharmacy, 3 Victor Babeş Street, 400012 Cluj-Napoca, Romania; parvualinaelena@yahoo.com; 4Faculty of Medicine, Iuliu Haţieganu University of Medicine and Pharmacy, 3 Victor Babeş Street, 400012 Cluj-Napoca, Romania; leandrosstoeber@yahoo.de; 5Alexandru-Borza Botanical Garden, Babeș-Bolyai University, 42 Republicii Street, 400015 Cluj-Napoca, Romania; casioana@yahoo.com; 6MEDFUTURE—Research Center for Advanced Medicine, “Iuliu-Hatieganu” University of Medicine and Pharmacy, 23 Marinescu Street, 400337 Cluj-Napoca, Romania

**Keywords:** onychomycosis, *Allium sativum*, antifungal, antioxidant

## Abstract

Onychomycosis is a major health problem due to its chronicity and resistance to therapy. Because some cases associate paronychia, any therapy must target the fungus and the inflammation. Medicinal plants represent an alternative for onychomycosis control. In the present work the antifungal and antioxidant activities of *Alium sativum* extract against *Meyerozyma guilliermondii* (Wick.) Kurtzman & M. Suzuki and *Rhodotorula mucilaginosa* (A. Jörg.) F.C. Harrison, isolated for the first time from a toenail onychomycosis case, were investigated. The fungal species were confirmed by DNA molecular analysis. *A. sativum* minimum inhibitory concentration (MIC) and ultrastructural effects were examined. At the MIC concentration (120 mg/mL) the micrographs indicated severe structural alterations with cell death. The antioxidant properties of the *A. sativum* extract were evaluated is a rat turpentine oil induced inflammation, and compared to an anti-inflammatory drug, diclofenac, and the main compound from the extract, allicin. *A. sativum* reduced serum total oxidative status, malondialdehyde and nitric oxide production, and increased total thiols. The effects were comparable to those of allicin and diclofenac. In conclusion, the garlic extract had antifungal effects against *M. guilliermondii* and *R. mucilaginosa*, and antioxidant effect in turpentine-induced inflammation. Together, the antifungal and antioxidant activities support that *A. sativum* is a potential alternative treatment in onychomycosis.

## 1. Introduction

Onychomycosis is a worldwide common nail fungal infection causing nail white or yellow discoloration, thickening, and separation from the nail bed [1]. Common pathogens in onychomycosis are dermatophytes, non-dermatophyte molds (NDM) and yeasts [2]. The International Society for Human and Animal Mycology proposed that the diagnosis of onychomycosis should be replaced by tinea unguium in infections with dermatophyte, by onyxis in infections with yeasts, by ungual candidiasis in infections with *Candida*, and by ungual mycosis when the infection is a NDM [3]. Tinea unguium are caused mostly by three genera *Trichophyton*, *Microsporum*, and *Epidermophyton*, and fungal infections by *Trichophyton rubrum*, *T*. *mentagrophytes*, and *Epidermophyton floccosum* are the most frequent worldwide. Among the yeasts *Candida* species are the most common [2]. Although *Candida albicans* is still the most frequent yeast pathogen [4], other *Candida* species such as *C. glabrata, C. krusei, C. tropicalis, C. parapsilosis* [5]*, C. guilliermondii* [6], etc. are proved to be important pathogens as well [7]. *Candida guilliermondii* has been isolated from numerous human infections, mostly of cutaneous origin, but it is also found on normal skin, in sea water, animal faeces, wasps, buttermilk, leather, fish, and beer [8]. The species *Meyerozyma guilliermondii* is most often associated with onychomycosis and candidemia [9].

Onychomycosis is a major health problem because it is so widespread and it is characterized by extreme chronicity and resistance to therapy [10]. For *Candida* infections treatment difficulties are connected to the relative genetic similarity between *Candida* and humans, which decreases the number of available targets. Many antifungals used for the treatment of *Candida* infections target the ergosterol from the fungal plasma membrane. Treatment-resistance of *Candida* is usually the consequence of ergosterol replacement by a precursor molecule, or a general reduction of the sterol in the plasma membrane.

Other yeasts recently accepted as the causative agents of onychomycosis belong to *Rhodotorula* genus: *Rhodotorula mucilaginosa* [11] and *R. minuta* [12]. In vitro susceptibility tests have shown that in onychomycosis the fungus *R. mucilaginosa* is susceptible to low concentrations of 5-flucytosine and amphotericin B, but resistant to high concentrations of terbinafine, voriconazole, itraconazole, and fluconazole [11].

An alternative for onychomycosis control may be the use of medicinal plants due to their richness in bioactive molecules. Among very important medicinal plants are *Allium* plants which are still intensively studied for their beneficial health effects [13]. *Allium sativum* is one of the most cultivated plants and it is used as a spice and for its therapeutical properties, such as antifungal effects [14].

Some cases of onychomycosis are associated with paronychia, a periungual inflammation [15,16]. Moreover, some studies suggested that low grade systemic inflammation may be associated with onychomycosis as well [17]. The inflammatory response involves an oxidative stress as an injury mechanism [18]. Because reactive oxygen species (ROS) can freely cross intracellular membranes, ROS may induce local and systemic consequences by damaging proteins, lipids, and DNA [19]. Furthermore, untreated onychomycosis could lead to a secondary infection [20]. Considering these facts, oxidative stress should also be considered a secondary target in the therapy of onychomycosis.

The first aim of the study was to evaluate the antifungal effect of *A. sativum* extract against *M. guilliermondii* (Wick.) Kurtzman & M. Suzuki and *R. mucilaginosa* (A. Jörg.) F.C. Harrison causing onychomycosis, by finding the minimal inhibitory concentration (MIC) and using transmission electron microscopy (TEM). *A sativum* L. (garlic) extract effect on the germination and the in vitro growth of the yeasts was compared to the commercial antifungal naftifine hydrochloride (10 mg/mL, active substance in Exoderil). The second aim was to evaluate the chemical composition and the antioxidant properties of the *A. sativum* extract in order to target inflammation associated oxidative stress if paronychia complicates onychomycosis. A phytochemical analysis of the *A. sativum* extract was also performed.

## 2. Results

### 2.1. Phytochemical Analysis of Garlic Extract

In the analysed garlic extract the concentration of alliin was 1410 µg/g and that of allicin was 380 µg/g. Regarding phenolics, only gentisic acid, chlorogenic acid, 4-hydrobenzoic acid, and *p*-coumaric acid were found above the limit of quantification (LOQ), but no flavonoid was detected. Further, total phenolic content (TPC) and total tiosulfinate content (TTC) were determined. The data showed that the studied extract contains much more thiosulfinates (443 µg/g allicin equivalents) compared with phenolic acids (15 µg/g gallic acid equivalents) in good agreement with the Principal Component Analysis (PCA) data (see Appendix A)

### 2.2. Onychomycosis Characteristics

A clinical case of toenail total dystrophic onychomycosis (Figure 1A) was due to a mixed infection associating *M. guilliermondii* and *R. mucilaginosa* was analyzed (Figure 1B).

The identification of the fungi *M. guilliermondii* and *R. mucilaginosa* was made by the cultural characteristics of the colonies obtained on Sabouraud Dextrose Agar (SDA), the *M. guilliermondii* colonies being mucoid and white, and the *R. mucilaginosa* colonies being mucoid and red respectively (Figure 1B). The morphological characteristics of the two species were obtained using scanning electron microscopy (SEM). For *M. guilliermondii* the SEM micrographs showed spherical to subspherical yeast-like cells, of 2.0–4.0 × 3.0–6.5 μm in size. Besides these, the fungus had branched and smooth pseudohyphae with dense verticils of blastoconidia (Figure 2A,B). *R. mucilaginosa* SEM showed only spheroidal to oval budding cells (2.5–6.5 × 6.5–14.0 μm) with carotenoid pigments and without the rudimentary formation of hyphae, and *R. mucilaginosa* cells were covered by visible mucilage (Figure 2C,D).

### 2.3. Molecular Confirmation

The confirmation of the species was performed by DNA molecular analysis. The obtained DNA sequences were run as queries against the BLAST-NCBI nucleotide database [21] and identically matched *M. guilliermondii* (Wick.) Kurtzman & M. Suzuki (GeneBank: MN273503) which is the synonym for *Candida guilliermondii* (Castell.) Langeron & Guerra, and *R. mucilaginosa* (A. Jörg.) F.C. Harrison (GenBank: MN273504). The percentage identity and the other alignment indices (Max score, Total score, and Query coverage) were the highest (100%) when the DNA sequences were compared to *M. guilliermondii* and *R. mucilaginosa* sequences present in the NCBI database, confirming that the isolated species have the most similar DNA sequences to the above-mentioned species.

### 2.4. The Antifungal Effect of A. sativum Extract

The garlic extract inhibited the germination and the growth of *M. guilliermondii* on SDA. After three days of incubation, the control Petri dishes had white to cream colonies, of 11–12 mm in size, whereas the colonies from the experimental variants with plant extract had small colonies in accord with the concentration of garlic extract. 

On SDA with 4% garlic extract, the colonies had 9–10 mm and those obtained on SDA with 8% extract had 3–4 mm. No colonies were identified on SDA with 12% (120 mg/mL) garlic extract. Regarding naftifine hydrochloride the maximum of inhibition of *M. guilliermondii* colonies was 55–56% and corresponded to the concentration of 5% (500 µg/mL). The inhibition of naftifine against *M. guilliermondii* was of 55–56% for the concentrations of 3%, 4% and 5% respectively (Figure 3, Table 1).

The *R. mucilaginosa* control colonies were mucoid and had a diameter of 12–13 mm after 3 days of incubation on SDA. These colonies had red color because the cells possess a carotenoid biosynthetic pathway. On SDA with garlic extract the *R. mucilaginosa* colonies had red color and a smaller diameter than the control colonies according to the plant extract concentration. At the 4% garlic extract, the colonies had 10–11 mm and those obtained on SDA with 8% extract had 7–8 mm. The inhibitory effect of garlic extract against *R. mucilaginosa* was 22.5% in the case of 4% concentration and 90% for 12% concentration. 

In the experimental variants with naftifine hydrochloride, the colonies had white color in comparison with the control. The inhibitory effect of naftifine hydrochloride against *R. mucilaginosa* was of 21% for 0.5% concentration and 90% for 2.5%. Antifungal susceptibility test performed by the agar dilution method showed that garlic extract had a MIC of 14% (140 mg/mL) against *R. mucilaginosa* isolate and the naftifine hydrochloride of 3% (300 µg/mL). The results demonstrated that naftifine hydrochloride affected the carotenoid biosynthetic pathway producing depigmentation of cells growing on SDA (Figure 4, Table 2). The antifungal effect of garlic extract against the yeasts was compared to naftifine. Based on MIC determination the *R. mucilaginosa* isolate was significantly more susceptible to the naftifine, in general, than the *M. guilliermondii* isolate.

### 2.5. Ultrastructural Changes Produced by A. sativum Extract

On the TEM micrographs of the *M. guilliermondii* control cells the cellular organelles such as the cellular wall, plasmalemma, vacuole, mitochondria, and nucleus respectively, were clearly visible (Figure 5A–D). The distinctive cell wall layers are the inner cell wall layer adjacent to the plasmalemma and the electron-dense external cell wall layer (Figure 5C). The MIC of garlic extract penetrated the cellular and the organelles membrane of *M. guilliermondii* and caused the alteration and precipitation of the cytoplasmic content in all treated cells (Figure 5E,F). Because of the cytoplasmic alterations, the nucleus and the cell organelles were destroyed, resulting in cell death.

Like previously described, the ultrastructural characteristics of *R. mucilaginosa* control cells included the electron-dense and lamellate cell wall, plasmalemma, endoplasmic reticulum, mitochondrion, lipid and glycogen accumulation in the cytoplasm, and spherical to ovoid nucleus. In the cytoplasm of young cells were found numerous small lipid granules of uniform size and in mature and senescent cells were found larger lipid granules with variable shapes generated by fusion (Figure 6A–D). The *A. sativum* extract at MIC caused irreversible ultrastructural changes in *R. mucilaginosa* treated cells determining the loss of the structural integrity and affecting the germination capacity. The alteration and the precipitation of the cytoplasmic content, destroyed nucleus and cell organelles resulted in cell death (Figure 6E,F).

### 2.6. Antioxidant Properties Evaluation

The in vitro antioxidant activity of the *A. sativum* tested extract was investigated using the most frequently used methods, the 2,2′-azino-bis-3-ethylbenzthiazoline-6-sulphonic acid (ABTS) and 1,1-diphenyl-2-picrylhydrazyl (DPPH) bleaching assays. The activity was higher when the ABST bleaching assay was used compared to the DDPH bleaching assay (Figure 7A).

In vivo antioxidant effect of *A. sativum* extract evaluated on a rat experimental acute inflammation model was performed by measuring systemic oxidative stress markers. The effects were compared to diclofenac, an anti-inflammatory drug, and to allicin, the main compound from phytochemical analysis of the tested extract. Important increases in serum Oxidative Stress Index (OSI) (*p* < 0.001) and Total Oxidative Status (TOS) (*p* < 0.001) with concomitant decrease in Total Antioxidant Response (TAR) (*p* < 0.01) were seen in the animals from the Inflammation group (INFLAM). *A. sativum* treatments resulted in a substantial decrease in the levels of TOS (*p* < 0.01) and OSI (*p* < 0.01) and no important changes in TAR (*p* < 0.05) for all tested dilutions. On TOS allicin (*p* < 0.01) had similar effect ass *A. sativum*, but diclofenac (*p* < 0.01) had lower inhibitory effects (Table 3).

Serum nitrites and nitrates (NOx) were substantially elevated in the INFL group (*p* < 0.001) as compared to the control rats. The treatment with *A. sativum* extracts 100% (*p* < 0.001) and 50% (*p* < 0.001) induced a very important reduction of NOx production, but *A. sativum* extract 25% was less efficient (*p* < 0.01). Allicin was also a good inhibitor of NOx (*p* < 0.01), and this effect was comparable to *A. sativum* 100% and 50%, and better than diclofenac (*p* < 0.01). There was no considerable statistical difference between *A. sativum* extract 25% and diclofenac in terms of NOx production (*p* > 0.05, Table 3). Total thiols (SH) were reduced in the INFLAM group (*p* < 0.001), and the treatments with allicin and diclofenac caused an important increase of SH (*p* < 0.001). *A. sativum* extract 50% had smaller stimulatory effect on the SH than the A. sativum extracts 100% and 25% (*p* < 0.001, Table 3).

Inflammation was associated with a considerable elevation of MDA (*p* < 0.001), and all *A. sativum* extract dilutions 25% reduced MDA production (*p* < 0.01). Allicin and diclofenac were equally efficient on MDA reduction (*p* < 0.01), and similar to *A. sativum* extract (Table 3).

## 3. Discussion

Our study presents for the first time a clinical case of toenail total dystrophic onychomycosis due to a mixed infection associating *M. guilliermondii* and *R. mucilaginosa*, and showed that the garlic extract had antifungal effect against the two species.

Onychomycosis is a multifactorial disease. Based on morphologic patterns and mode of nail invasion it was divided into several classes [23]: distal and lateral subungual onychomycosis; proximal subungual onychomycosis; superficial onychomycosis; total dystrophic onychomycosis; endonyx subungual onychomycosis, which is rare. Some nails have features from a combination of classes [1]. Usually the main causative agent of onychomycosis is only one fungus [2], but mixed infections of dermatophytes and NDM or of dermatophytes and yeasts [10] in individuals with this disease were reported too. Since onychomycosis is caused by different fungal species, their precise and correct identification is very important in order to select the most appropriate antifungal therapy [11].

In our study, the identification of the fungi *M. guilliermondii* and *R. mucilaginosa* was made by morphological and cultural characteristics of the colonies obtained on SDA and confirmed by DNA molecular analysis. It is known that *Allium* plants are rich in active compounds such as alliin, allicin, ajoene, sterols, flavonoids, and phenolic acids [24]. The phytochemical analysis of the *A. sativum* extract found an important content of alliin and allicin. Besides the sulphur-containing compounds, garlic is also known to contain some phenolic compounds, mostly in the form of phenolic acids [25] known to have antioxidant and anti-inflammatory properties [24]. The chromatographic method employed led to the identification of only four phenolic acids and no flavonoid, even though other *Allium* species are known to contain high amounts of flavonoids [26].

Allicin is one of the most important bioactive compounds from *Allium* species that exhibits antioxidant and antifungal, antibacterial, antiviral, antiparasitic activity and is effective against many fungal species [27]. The garlic extract inhibited the germination and the growth of *M. guilliermondii* and *R. mucilaginosa* on SDA. Our results are in accord with literature data regarding antifungal effect of *A. sativum* extract against human pathogenic fungi like *Candida* species [4], *Trichophyton mentagrophytes*, *T. rubrum*, *T. verrucosum*, *Microsporum gypseum* and *Epidermophyton floccosum* etc. Therefore, garlic extract exhibited a fungicidal effect against *Candida albicans*, *C. tropicalis* and *C. krusei* at the concentration of 100 mg/mL. The extract fully suppressed the production of hyphae by *Candida albicans* even at the lowest concentration tested, of 20 mg/mL. On the other hand, the MIC of garlic oil against *C. albicans* was 0.35 μg/mL [4].

The effect of garlic extract against *R. mucilaginosa* isolate causing onychomycosis is not mentioned in the literature. For treatment of *Rhodotorula* infections different chemicals and plant extracts (*Punica granatum*, *Salvia officinalis*, *Laurus nobilis*, and *Thymbra spicata*) were tested [28].

The garlic extract affected the fungal cell wall, which is a complex and dynamic structure, a contact surface with the host environment, but in the same time antifungal treatment target [5]. The *A. sativum* extract had fungicidal effect by causing irreversible ultrastructural changes in *M. guilliermondii* and *R. mucilaginosa* cells, which determined loss of structural integrity and affected the germination capacity. The alteration and precipitation of the cytoplasmic content, destroyed nucleus and cell organelles resulted in cell death. Our results are in accord with those obtained in the treatment of *Candida albicans* with garlic oil in MIC of 0.35 μg/mL [4].

The ultrastructural changes induced by the *A. sativum* extract may be correlated to the antifungal compounds, recommending the extract as an alternative in the treatment of dermatophytic fungi, *Candida* isolates [4]. Generally, efficient antioxidant plant extracts contain compounds that also display antimicrobial activity [29]. By using plant extracts, rich in natural antioxidants, oxidative stress can be reduced. The in vitro antioxidant activity of the tested extract was higher when ABST bleaching assay was used compared to DDPH bleaching assay. This indicates that a mechanism of electron transfer predominates compared to hydrogen atom transfer, suggesting a higher contribution of sulphur-containing agents (i.e., thiosulfinates, sulphides, etc.) rather than phenolic compounds which usually act as hydrogen atom transfer. Our phytochemical analysis confirmed the higher allin and allicin contents of the *A. sativum* extract.

Because it was found that some extracts exhibit antioxidant activity both in vitro and *in vivo*, but for other extracts, the in vitro antioxidant activity does not apply to in vivo models, we further tested the in vivo antioxidant effects. Furthermore, it was known that in vitro and in vivo antioxidant effects may not correspond because polyphenols may also act as prooxidants via Fenton reaction [24].

*In vivo,* the *A. sativum* extract effect consisted in oxidant reduction. Oxidant reduction was evidentiated by measuring TOS, NO and MDA. Endogenous nitric oxide (NO) is synthesized from L-arginine by three major nitric oxide synthases: endothelial (eNOS), neural (nNOS), and inducible nitric oxide synthases (iNOS). NO produced by iNOS plays an important role in host defense. High level or prolonged induction of NO may lead to pathological consequences in inflammatory diseases due to the involvement in the oxidative stress. Reduction of serum NOx by *A. sativum* extracts was important, and followed the same pattern like TOS and OSI. MDA is a secondary product of lipid peroxidation. The inhibitory effect of the *A. sativum* extract on MDA production also shows that the extract is a good antioxidant.

Because the main compound of the extract was allicin, *A. sativum* extract effects were compared to allicin. Furthermore, because the antioxidant effect is anti-inflammatory, *A. sativum* activity was also compared to a nonsteroidal anti-inflammatory drug, diclofenac. Finally, it was found that *A. sativum* antioxidant activity was better or comparable with those of allicin and diclofenac. Considering that local or systemic inflammation may associate to onychomycosis, and oxidative stress is an injury mechanism in inflammation, *A. sativum* extracts antioxidant activity may recommend this product for local or systemic anti-inflammatory therapy in onychomycosis.

## 4. Materials and Methods

### 4.1. Chemicals

The chemicals used were naftifine (active substance in Exoderil (Sandoz GmbH, (Kundl, Austria)), Sabouraud dextrose agar (SDA), diclofenac (Sigma-Aldrich, (St. Louis, MO, USA)) and allicin (described in the Appendix A). Below it says this was purchased from Aldrich.

### 4.2. Allium sativum Extract

Garlic (*Allium sativum* L.) bulbs were collected from a private garden of Cluj-Napoca. A voucher specimen (CL 666212) is deposited at the Herbarium of Babeş-Bolyai University, Cluj-Napoca, Romania. Fresh *A. sativum* bulb fragments of 0.5–1.0 cm were extracted with 60% ethanol (Merck, Bucureşti, Romania) by a cold repercolation method, at room temperature, for 3 days. The *A. sativum* extract, containing 1 g plant material in 1 mL of 20% ethanol (*w:v/g:mL*) was obtained by filtration.

### 4.3. Phytochemical Analysis

Two HPLC-DAD-MS protocols were optimized to separate and determine the main phytoconstituents of the studied extract; the first HPLC protocol was optimized for allicin and alliin determination and the second one for polyphenolic compounds determination. Total phenolic content (TPC) was determined using Folin Ciocalteu reagent and expressed in gallic acid equivalents as previously described [30]. Total thiosulfinate content (TTC) was determined using a kinetic assay based on 4-mercaptopyridine bleaching assay monitored at 324 nm as described by Miron et al. and was expressed in allicin equivalents based on a calibration curved constructed using this standard [22]. More information may be found in the Appendix A.

### 4.4. Fungal Strain Isolation and Growth Conditions

The fungi *R. mucilaginosa* (A. Jörg.) F.C. Harrison and *M. guilliermondii* (Wick.) Kurtzman & M. Suzuki (syn. *Candida guilliermondii* (Castell.) Langeron & Guerra) were isolated from toenails affected by onychomycosis obtained from a patient woman of 88 years with chronic HBV hepatitis. After nail asepsis with 70% ethanol, the distal and lateral fragments of both toenail plates with subungual debris were removed with a sterile nail clipper. 

The Ethics Committee of the Iuliu Haţieganu University of Medicine and Pharmacy, Cluj-Napoca (Romania) approved the study and informed written consent were obtained from the patient before enrolling her in the study. All experiments in this study were performed in accordance with relevant guidelines and regulations.

The *R. mucilaginosa* and *M. guilliermondii* fungal isolates were obtained using 2–3 mm toenail fragments disinfected in 20% ethanol for 1 min and inoculated onto SDA control media in Petri dishes. The fungal samples were identified by morphological and cultural characteristics of the colonies obtained by the standard method of triple culture (insemination of the inoculum in three points on the surface of SDA in Petri dishes) and after incubation at 22 °C for 3 days. All experiments were run three times [31].

### 4.5. Molecular Confirmation of the Fungi

The species *M. guilliermondii* and *R. mucilaginosa,* isolated on SDA from a human toenail affected by onychomycosis were molecularly confirmed targeting the Internal Transcribed Spacer using the ITS1 and ITS2 primers [32]. The DNA extraction was performed using the commercial kit Animal and Fungi DNA Preparation Kit^®^ (Jena Bioscience, Jena, Germany) according to manufacturers’ protocol. The PCR (Polymerase chain reaction) amplification took place in a 25 µL final volume of mixture reaction containing: 1X MangoTaq Colored Reaction Buffer (Bioline, London, UK), 2.5 mM MgCl_2_ (Bioline), 0.5 mM dNTP (Bioline), 0.5 mM of each primer (Macrogen Inc., Seoul, South Korea), 1.25U/µL MangoTaq (Bioline) and 2 µL of DNA. The amplification series consisted of 35 cycles of the following: 95 °C for 30 sec, 56 °C for 30 sec and 72 °C for 30 sec.

### 4.6. The Antifungal Activity Evaluation

In order to determine the effect of *A. sativum* extract, by agar dilution method, fungal cells were inoculated on SDA with different concentrations of garlic extract: 2%, 4%, 6%, 8%, 10%, and 12% in case of *M. guilliermondii* and 2%, 4%, 6%, 8%, 10%, 12%, and 14% for *R. mucilaginosa*. The negative control was without extract (nutritive medium and 20% ethanol). An antifungal control with naftifine (Exoderil Sandoz GmbH Kundl, Austria -10 mg of naftifine hydrochloride/mL) was also used. In case of naftifine, experimental variants with 0.5%, 1%, 2%, 3%, 4%, and 5% for *M. guilliermondii* and 0.1%, 0.5%, 1%, 1.5%, 2%, 2.5%, and 3% for *R. mucilaginosa* were performed. The antifungal activity of the *A. sativum* extract was expressed as MIC. The percentage of mycelial growth inhibition (*P*) at each concentration was calculated using the formula *P = (C−T) × 100/C*, where *C* is the diameter of the control colony and *T* is the diameter of the treated colony [31].

### 4.7. Electron Microscopy

The morphology of the *R. mucilaginosa* and *M. guilliermondii* control cells was analyzed by scanning electron microscopy (SEM) using a JSM 5510 LV instrument (JEOL Co., Tokyo, Japan). The ultrastructure of the *R. mucilaginosa* and *M. guilliermondii* control cells and those treated for 1 h with the MIC of garlic extract were analyzed by TEM using a JEOL JEM 1010 electron microscope.

The fungal samples for electron microscopy were prepared according to literature [31] and the chemicals used were glutaraldehyde, resin (Epon 812), lead citrate, uranyl acetate, bismuth subnitrate (Electron Microscopy Sciences, Fort Washington, MD, USA), sticky carbon tabs, colloidal carbon coated grids (Agar Scientific, Cambridge, UK). Details regarding the sample work protocol for electron microscopy were presented in other reports [31].

### 4.8. In Vitro Antioxidant Effect

Antioxidant activity was evaluated using DPPH and ABTS radicals scavenging assays according to the protocols previously published [33] with the only difference that the results were expressed this time in quercetin equivalents obtained from calibration curves using quercetin as a standard (R^2^ > 0.995) for both assays.

### 4.9. In Vivo Antioxidant Effect

The experiments were carried out on male Albino Wistar rats, weighing 200–250 g, that were bred in the Animal Facility of Iuliu Haţieganu University of Medicine and Pharmacy, Cluj-Napoca. The animals were housed in controlled conditions (12 h light/dark cycle, at an average temperature of 21–22 °C), and had free access to standard pellet diet (Cantacuzino Institute, Bucharest, Romania) and water *ad libitum*. The animals were randomly assigned to seven groups (*n* = 5). *A. sativum* extract was administrated orally by gavage (1 mL/animal) in three dilutions (100%, 50%, and 25%) for seven days. Rats from the negative control group (CONTROL) and from the inflammation group (INFLAM) received by gavage for seven days tap water (1mL/animal). For seven days an anti-inflammatory control treated by gavage with diclofenac (DICLOFENAC) (10 mg/kg body-weight (bw) for seven days, and a group treated by gavage with 1mL/animal of allicin solution (450 µg/mL) were also used [34]. Allicin was purchased from Sigma-Aldrich (St. Louis, MO, USA) and diluted in water. The allicin dose corresponded to allicin concentration from one mL of undiluted *A. sativum* extract. In the 8th day, inflammation was induced by turpentine oil (i.m. 0,6 mL/kg bw) in the animals treated with the extracts, in INFLAM, DICLOFENAC and allicin groups [19]. CONTROL animals were injected with 0.9% saline (i.m. 0.6 mL/kg bw). Twenty-four hours after the inflammation induction, the rats were anesthetized using ketamine (60 mg/kg bw) and xylazine (15 mg/kg bw), blood was withdrawn by retro-orbital puncture, and serum was stored at −80 °C until use. The in vivo experiments were performed in triplicate (Table 4). At the end of the experiment under general anesthesia, animals were killed by cervical dislocation [19].

All treatments that involved animals were rigorously in accordance with EU Directive 2010/63/EU (European guidelines and rules). The Research Ethics Committee from Iuliu Haţieganu University of Medicine and Pharmacy Cluj-Napoca approved the research protocol (22/2016).

### 4.10. Systemic Oxidative Stress Markers Determination

The TOS of the serum was measured by a colorimetric assay and the results are expressed in μmol H_2_O_2_ equiv/L. Serum total antioxidant response (TAR) was measured by using a colorimetric assay, and results are expressed as mmol Trolox equiv/L. The oxidative stress index (OSI), an indicator of the degree of oxidative stress was calculated: OSI (arbitrary unit) =TOS (mol H_2_O_2_ equiv/L)/TAR(mmol Trolox equiv/L) [35].

The Griess reaction was used to indirectly determine nitric oxide synthesis by measuring NOx and the results were expressed as nitrite μmol/L [35]. Malondialdehyde (MDA), a lipid peroxidation marker, was measured using thiobarbituric acid and the results were expressed as nmol/mL of serum. Total thiols (SH) were estimated using Ellman’s reagent and the results were expressed as mmol GSH/mL [36]. All serum spectrophotometric measurements were performed using a Jasco V-530 UV-Vis spectrophotometer (Jasco International Co. Ltd., Tokyo, Japan).

### 4.11. Statistical Analysis

Statistical analyses were performed using the program R environment, version 2.14.1. The results for each group were expressed as mean ± SD. Data were evaluated by analysis of variance (ANOVA) followed by Bonferroni-Holm posthoc test. A *p* value of ≤0.05 was considered statistically significant. The correlation analysis was performed by the Pearson test.

## 5. Conclusions

The findings of this study have shown for the first time that toenail onychomycosis may be determined by a mixed infection associating the yeasts *M. guilliermondii* and *R. mucilaginosa* and completed the list of aetiologic agents of this disease. Since the *A. sativum* extract had antifungal effect against *M. guilliermondii* and *R. mucilaginosa* isolates causing onychomycosis, this extract can be an attractive or a complementary alternative to common antifungal drugs. Due to the antioxidant effect, *A. sativum* extracts may target in the same time the oxidative stress from the inflammation associated to onychomycosis. The mechanisms of action of garlic constituents in vivo still require much research.

### Data Availability

The datasets generated during and/or analyzed during the current study are available from the corresponding author on reasonable request.

## Figures and Tables

**Figure 1 molecules-24-03958-f001:**
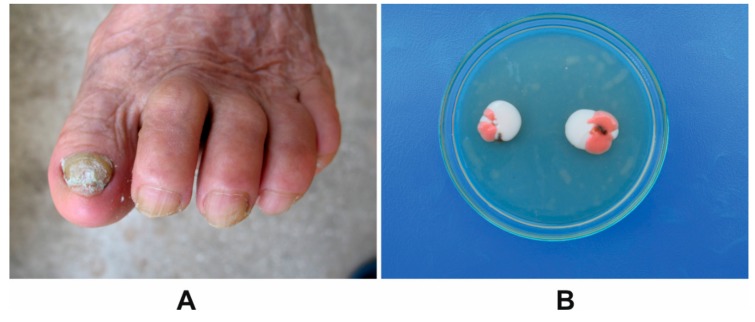
(**A**) Toenails affected by onychomycosis; (**B**) Colonies of *M. guilliermondii* (white to cream) and *R. mucilaginosa* (in red) around a toenail fragment on SDA.

**Figure 2 molecules-24-03958-f002:**
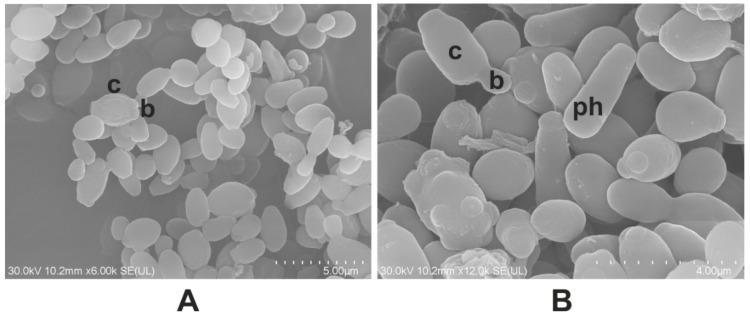
Scanning electron micrographs showing *M. guillermondii* (**A**,**B**) and *R. mucilaginosa* (**C**,**D**). Legend: b, bud; c, cell; m, mucilage; ph, pseudohyphae.

**Figure 3 molecules-24-03958-f003:**
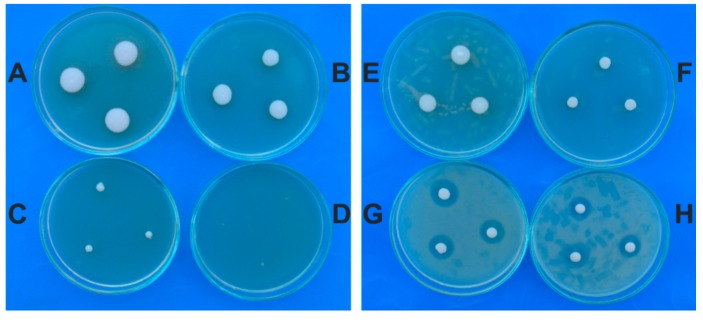
Colonies of *M. guilliermondii* on SDA: control (**A**); with garlic extract 4% (**B**); with garlic extract 8% (**C**); with garlic extract 12% (without colony) (**D**); with naftifine 0,5% (**E**); with naftifine 1% (**F**); with naftifine 3% (**G**); with naftifine 5% (**H**).

**Figure 4 molecules-24-03958-f004:**
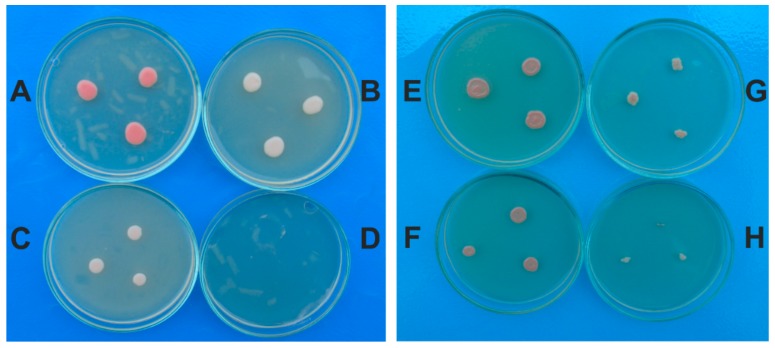
Colonies of *R. mucilaginosa* on SDA: with garlic extract 4% (**A**); with garlic extract 6% (**B**); with garlic extract 8% (**C**); with garlic extract 10% (**D**); control (**E**); with naftifine 0.1% (**F**); with naftifine 1.5% (**G**); with naftifine 3% (without colony) (**H**).

**Figure 5 molecules-24-03958-f005:**
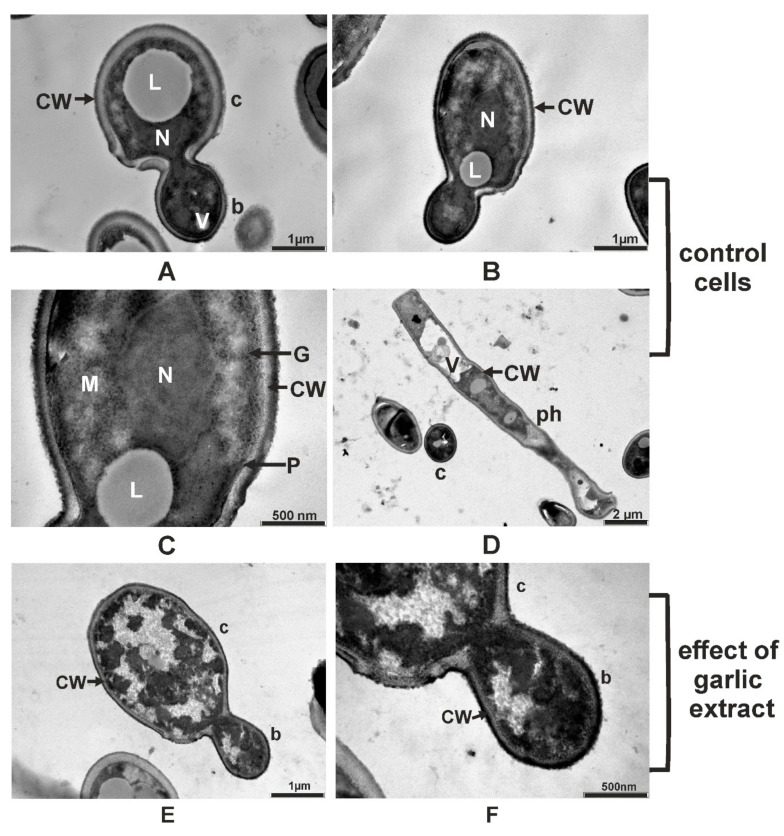
Transmission electron micrographs of *M. guilliermondii* showing ultrastructural components of control cells: longitudinal section of bud and cell (**A,B**); longitudinal section of cell (detailed) (**C**); longitudinal section of pseudohyphae (ph) and cross section of cells (**D**), and the changes produced by garlic extract in 12% minimum inhibitory concentration: longitudinal section of cell (**E**) longitudinal section of cell (detailed) (**F**). Legend: b, bud; c, cell; CW, cell wall; G, glycogen; L, lipid; M, mitochondrion; N, nucleus; P, plasmalemma; V, vacuole.

**Figure 6 molecules-24-03958-f006:**
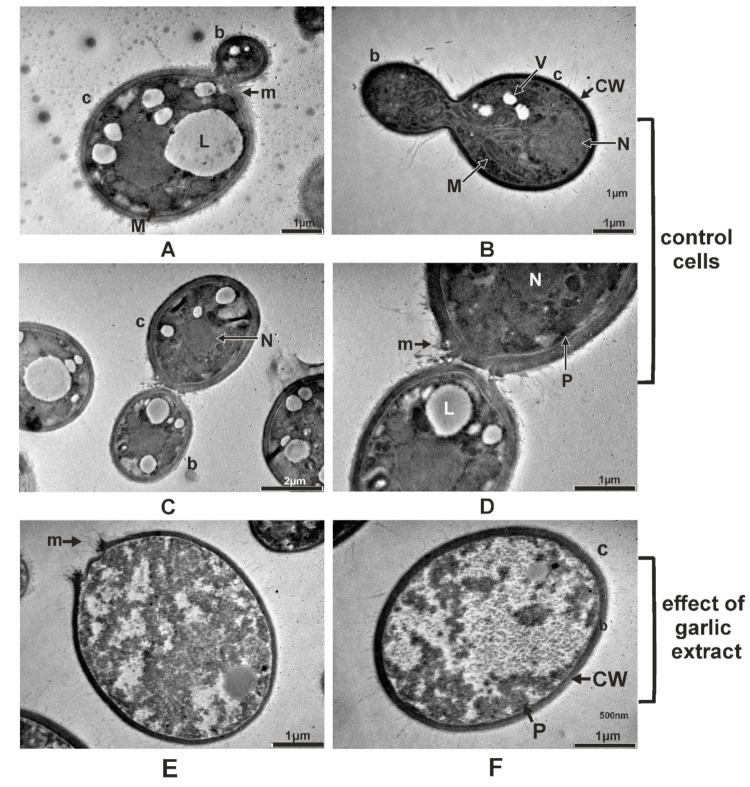
Transmission electron micrographs of *R. mucilaginosa* showing ultrastructural components of control cells: longitudinal section of bud and cell (**A,B,C**); longitudinal section of cell and bud (detailed) (**D**), and the changes produced by garlic extract in 14% minimum inhibitory concentration: longitudinal section of cell (**E**), cross section of cell (**F**). Legend: b, bud; c, cell; CW, cell wall; ER, endoplasmic reticulum; G, glycogen; L, lipid; M, mitochondrion; m, mucilage; N, nucleus; P, plasmalemma; V, vacuole.

**Figure 7 molecules-24-03958-f007:**
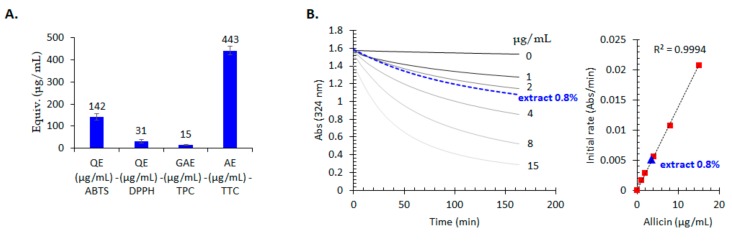
(**A**) In vitro antioxidant activity evaluated using ABTS and DPPH scavenging assays and expressed as quercetin equivalents (QE), total phenolic content (TPC) determined using Folin-Ciolalteau reagent and expressed as gallic acid equivalents (GAE) and total thiosulfinate content (TTC) and expressed in allicin equivalents (AE), *n* = 3, error bars are standard deviations (**B**) Kinetic determination of total thiosulfinate content of the studied *A. sativum* extract (0.8%) using Miron et al. [22] kinetic assay and the allicin standards (0–15 μg/mL) and the corresponding calibration curve.

**Table 1 molecules-24-03958-t001:** In vitro effect of *A. sativum* extract and naftifine against *M. guilliermondii* isolate using the agar dilution method.

Garlic Extract Concentration (%)	Colony ^a^ Diameter (mm)	P ^a^	Naftifine Solution Concentration (%)	Colony ^b^ Diameter (mm)	P ^b^
C	11.66 ± 0.81	0	C	11.66 ± 0.81	0
2	11.16 ± 0.40	4.28 ± 0.40	0.5	10.0 ± 0.63	14.23 ± 0.63
4	9.66 ± 0.51	17.15 ± 0.51	1	8.33 ± 0.51	28.55 ± 0.51
6	6.66 ± 0.51	42.88 ± 0.51	2	6.83 ± 0.83	41.42 ± 0.83
8	3.50 ± 0.54	69.98 ± 0.54	3	5.16 ± 0.40	55.74 ± 0.40
10	1.33 ± 0.51	88.59 ± 0.51	4	5.16 ± 0.40	55.74 ± 0.40
12	0	100	5	5.16 ± 0.40	55.74 ± 0.40

^a^ the effect of *A. sativum* extract; ^b^ the effect of naftifine solution; C control (20% aq. EtOH); P mycelial growth inhibition - results are the mean ± SD of six experiments.

**Table 2 molecules-24-03958-t002:** In vitro effect of *A. sativum* extract and naftifine against *R. mucilaginosa* isolate using the agar dilution method.

Garlic Extract Concentration (%)	Colony ^a^ Diameter (mm)	P ^a^	Naftifine Solution Concentration (%)	Colony ^b^ Diameter (mm)	P ^b^
C	13.33 ± 0.51	0	C	13.33 ± 0.51	0
2	12.16 ± 0.75	8.77 ± 0.75	0.1	12.83 ± 0.40	3.75 ± 0.40
4	10.33 ± 0.51	22.50 ± 0.51	0.5	10.5 ± 0.54	21.23 ± 0.54
6	9.0 ± 0.63	32.48 ± 0.63	1.0	7.83 ± 0.40	41.26 ± 0.40
8	7.33 ± 0.57	45.01 ± 0.57	1.5	6.33 ± 0.51	52.51 ± 0.51
10	4.0 ± 0.63	69.99 ± 0.63	2	3.66 ± 0.81	72.54 ± 0.81
12	1.33 ± 0.51	90.02 ± 0.51	2.5	1.33 ± 0.51	90.02 ± 0.51
14		100	3.0	0	100

^a^ the effect of *A. sativum* extract; ^b^ the effect of naftifine solution; C control (20% aq. EtOH); P = mycelial growth inhibition - results are the mean ± SD of 6 experiments.

**Table 3 molecules-24-03958-t003:** In vivo serum oxidative stress markers.

	TOS (μmol H_2_O_2_ equiv/L)	TAR (mmol trolox equiv/L)	OSI	NOx (μmol/L)	MDA (nmol/mL)	SH mmol (TSH/mL)
Control	38.22 ± 4.77 a***	1.0897 ± 0.0014 a**	35.04 ± 4.39 a***	52.88 ± 2.60 a***	5.82 ± 0.56 a***	0.67 ± 0.07 a***
Inflammation	66.35 ± 8.49 b**, c*	1.0891 ± 0.0009 b*, c*	60.90 ± 7.81 b**, c*	82.42 ± 0.27 b***, c**	7.50 ± 0.77 b***, c*	0.50 ± 0.08 b**, c*
Diclofenac	48.65 ± 8.36 a**, b*	1.0970 ± 0.0017 b*	44.47 ± 7.59 a***, b*	58.71 ± 5.29 a**, b*	5.75 ± 0.80 a**, b*	0.71 ± 0.11 a**, b*
Allicin	33.76 ± 4.82 a**, c*	1.0884 ± 0.0005 a*, c*	31.02 ± 4.43 a**, c*	40.30 ± 6.12 a***, c*	5.57 ± 0.48 a**, c*	0.75 ± 0.13 a**, c*
*A. sativum* 100%	33.13 ± 8.06 a**, b*, c*	1.0894 ± 0.0009 a*, b*, c*	30.43 ± 7.39 a**, b*, c*	38.12 ± 7.57 a***, b*, c**	5.54 ± 0.32 a**, b*, c*	0.60 ± 0.09 a**, b*, c*
*A. sativum* 50%	31.88 ± 6.91 a**, b*, c*	1.0884 ± 0.0009 a*, b*, c*	29.27 ± 6.34 a**, b*, c*	38.15 ± 8.10 a***, b*, c**	5.48 ± 0.63 a**, b*, c*	0.81 ± 0.29 a***, b*, c*
*A. sativum* 25%	32.25 ± 3.75 a**, b*, c*	1.0884 ± 0.0005 a*, b*, c*	29.63 ± 3.43 a**, b*, c*	53.91 ± 8.83 a**,b*, c*	5.79 ± 0.51 a**, b*, c*	0.62 ± 0.15 a**, b*, c*

Values are expressed as average ± standard deviation (SD) (*n* = 5). TOS: total oxidative status; TAR: total antioxidant reactivity; OSI: oxidative stress index; NOx: nitrites and nitrates; SH: total thiols; MDA: malondialdehyde, SH total thiols. a vs INFLAM; b vs Allicin; c vs DICLOFENAC; *p* > 0.05 *; *p* < 0.01 **; *p* < 0.001 ***.

**Table 4 molecules-24-03958-t004:**
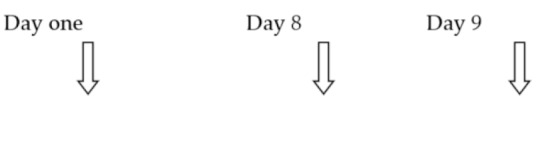
Flowchart of the in vivo experiment.

Group	7 Days Treatment by Gavage: 1mL/Animal	i.m. 0.6 mL/kg bw	
Control	tap water	saline solution	Blood collection under general anesthesia
Inflammation	tap water	turpentine oil	
Diclofenac	10 mg/kg bw	turpentine oil	
Allicin	450 μg/mL	turpentine oil	
*A. sativum* 100%	1mL/animal	turpentine oil	
*A. sativum* 50%	1mL/animal	turpentine oil	
*A. sativum* 25%	1mL/animal	turpentine oil	

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
