# Peer review of "Allium sativum Extract Chemical Composition, Antioxidant Activity and Antifungal Effect against Meyerozyma guilliermondii and Rhodotorula mucilaginosa Causing Onychomycosis"

_molecules, 2019, doi:10.3390/molecules24213958_

Round 1

Reviewer 1 Report

In the  paper entitled “ Allium sativum extract chemical composition, antioxidant activity and antifungal effect against Meyerozyma guilliermondii and Rhodotorula mucilaginosa causing onychomycosis” the authors investigated in vitro the antifungal effect of  Allium sativum extract as well as its antioxidant properties in vitro and in vivo. Moreover, the authors studied the chemical composition of this extract.  I have read the manuscript and I think it should be thoroughly reorganized.

In my opinion, the authors in their work should focus on the antifungal properties of Allium extract and its antioxidant properties. The composition of Allium extract has been studied by various researchers and has low novelty value. I strongly suggest attach the results connected with Allium extract composition in the Supplementary material. In this way, the paper will be more transparent.

The abstract should contain the background, the aim of the study, results and conclusions.

At the end of the introduction, the aim of the study should be redefined (evaluation of chemical composition is not the aim in my opinion). Moreover, it should be explained, why the antioxidant effects of Allium extract was studied in relation to its antifungal effect.

In the text and in the results presented in Table 1, different units were used – 1410 μg/mL vs. μg/g. In Table 1 and 2 different units of LOD were used. In the legend of Table 2 abbreviation LOQ is used, however it does not appear anywhere else.

Why does the Figure 3A present in vitro antioxidant activity of garlic extract together with kinetic determination of total thiosulfinate content?

What does SDA mean?

DNA molecular analysis may be a separate paragraph and more details can be provided.

Figure 6 and 7 – please, use a consistent order in both figures (i.e. control, naftifine and garlic extract or control, garlic extract and naftifine).

The naftifine was used as a reference compound, so its effect on colonies should not be discussed separately.

Figure 8 and 9 – parts A-D present components of control cells, while E and F present effects of garlic extract – please insert these information  into pictures. 

The abbreviation MIC is not explained.

The abbreviations once introduced, should be consistently used in the text.

In the results section, more details about antioxidant properties in in vitro studies should be provided.

Table 5 – it is completely chaotic: TAR, not TAC; abbreviations: INFL or INFLAM, DICLO or DICLOFENAC. In my opinion, INFL should be inserted just after CONTROL and then A. sativum and other groups. Why TAR has exactly the same value in all studied groups? Is it possible? Units: equiv/L not mM or microM/L! MDA – in the methods, the unit nmol/ml of serum is used and in the Table there is nM/L! What was measured as “total thiols”– only non-protein thiols or total thiols (protein and non-protein)? If total thiols (TSH) were determined, they included not only GSH but also protein thiols. Why was not statistical analysis performed on allicin group? Please, explain, why allicin group was studied.

In the results section, the detailed values of p (i.e. p=1.22E-06) are not necessary, please give more general information (i.e. p<0.0001).

In the results on MDA, it is not true that A. sativum dilutions of 50% and 25% reduced MDA more than the undiluted extract 100% (100% extract – 5.54, while 25% - 5,79).

Experimental design of the in vivo studies should be presented in a diagram. In the present form it is not clear and difficult to understand. There is no information about the dose of allicin. How were allicin and diclofenac administered to animals? What was a solvent for allicin and what for garlic extract? What does it mean “The experiments were performed in triplicates” in reference to animals groups?

The antifungal effect of A. sativum extract – if 12% extract contains 120 mg /ml, why 5%  contains 500 μg/ml. I think it is a mistake.

Please, try to limit the number of references – 50 are too many in my opinion for experimental paper.

Language correction is suggested.

Author Response

Tamara Milicevic

Assistant Editor

MDPI DOO

                                                                                                            October, 2019

Dear Mrs. Milicevic,

Subject: Submission of revised paper Molecules-620774

Thank you for the October 15th, 2019 e-mail enclosing the reviewer’s comments.

We would like to thank the reviewers for the careful and thorough reading of this manuscript and for the thoughtful comments and constructive suggestions. We have carefully reviewed the comments (shown below in italic) and have revised the manuscript accordingly. Our responses are given in a point-by-point manner below (dark blue text). Changes to the manuscript are shown in underline.

We look forward to hearing from you regarding our submission. We would be glad to respond to any further questions and comments that you may have.

Sincerely,

Response to Reviewer #1

In my opinion, the authors in their work should focus on the antifungal properties of Allium extract and its antioxidant properties. The composition of Allium extract has been studied by various researchers and has low novelty value. I strongly suggest attach the results connected with Allium extract composition in the Supplementary material. In this way, the paper will be more transparent.

We performed the phytochemical analysis because there are changes owing to the cultivar and location of growth. Then we needed the phytochemical analysis in order to find which the main compound is, because we wanted to compare the effects of Allium sativum L. extract to the main compound effects administrated in a dose comparable to the one from the extract. We moved the detailed phytochemical analysis in the Supplementary material as suggested.

The abstract should contain the background, the aim of the study, results and conclusions.

We rephrased the abstract as follows:

Line 23: “Onychomycosis is a major health problem due to its chronicity and resistance to therapy.  Because some cases associate paronychia, therapy must target the fungus and the inflammation. An alternative for onychomycosis control may be medicinal plants. In the present work the antifungal and antioxidant activities of Alium sativum extract against Meyerozyma guilliermondii (Wick.) Kurtzman & M. Suzuki and Rhodotorula mucilaginosa (A. Jörg.) isolated for the first time from a toenail onychomycosis case were investigated. Fungal species were confirmed by DNA molecular analysis. A. sativum minimum inhibitory concentration (MIC) and ultrastructural effects were examined. At the MIC concentration (120 mg/mL) the micrographs indicated severe structural alterations with cell death. The antioxidant properties of the A. sativum extract were evaluated is a rat turpentine oil induced inflammation, and compared to an anti-inflammatory drug, diclofenac, and the main compound from the extract, allicin. A. sativum reduced serum total oxidative status, malondialdehyde and nitric oxide production, and increased total thiols. The effects were comparable to those of allicin and diclofenac. In conclusion, the garlic extract had antifungal effects against M. guilliermondii and R. mucilaginosa, and antioxidant effect in turpentine-induced inflammation. Together, the antifungal and antioxidant activities support that A. sativum is a potential alternative treatment in onychomycosis.

Keywords: onychomycosis, Allium sativum, antifungal, antioxidant”

At the end of the introduction, the aim of the study should be redefined (evaluation of chemical composition is not the aim in my opinion). Moreover, it should be explained, why the antioxidant effects of Allium extract was studied in relation to its antifungal effect.

We reformulated the study aims and explained why the antioxidant effects are in relation to its antifungal effect as follows:

Line 105: „The first aim of the study was to evaluate the antifungal effect of A. sativum extract against M. guilliermondii (Wick.) Kurtzman & M. Suzuki and R. mucilaginosa (A. Jörg.) F.C. Harrison causing onychomycosis, by finding the minimal inhibitory concentration (MIC) and using electron microscopy. The second aim was to evaluate the antioxidant properties of the A. sativum extract in order to target inflammation-associated oxidative stress if paronychia complicates onychomycosis. A phytochemical analysis of the A. sativum extract was also performed.”

In the text and in the results presented in Table 1, different units were used – 1410 μg/mL vs. μg/g. In Table 1 and 2 different units of LOD were used. In the legend of Table 2 abbreviation LOQ is used, however it does not appear anywhere else.

We corrected the units of measurement, now it is ug/g in the text as well as in Table 1 (current Table S1), as it should be. We removed LOQ which was not added since it is known to be 3xLOD.

Why does the Figure 3A present in vitro antioxidant activity of garlic extract together with kinetic determination of total thiosulfinate content?

Figure 3A (current Figure 7A) contains the results from two methods that were used to determined antioxidant activity (DPPH and ABTS bleaching), the results for total phenolics (via Folin-Ciocalteau reagent) and the total thiosulfinates contents (determined via the kinetic profiles from Figure 3B via external calibration with allicin). We consider appropriate this mode of presentation since it presents, besides the antioxidant activity also two main classes of phytocompounds from the garlic: phenolics and thiosulfinates. Therefore, we showed that the phenolics are a minor group of phytocompounds compared with thiosulfinates and these could be responsible for the antioxidant activity not phenolics as it is frequently the case in other plant extracts.

What does SDA mean?

SDA means Sabouraud Dextrose Agar. We defined the abbreviation at its first appearance in the text.

DNA molecular analysis may be a separate paragraph and more details can be provided.

The molecular analysis was indicated in the text in a separate paragraph and more details were provided. We added the following text.

Line 425: “The DNA extraction was performed using the commercial kit Animal and Fungi DNA Preparation Kit® (Jena Bioscience, Jena, Germany) according to manufacturers’ protocol. The PCR (Polymerase chain reaction) amplification took place in a 25µl final volume of mixture reaction containing: 1X MangoTaq Colored Reaction Buffer (Bioline, London, UK), 2.5mM MgCl2 (Bioline), 0.5 mM dNTP (Bioline, London, UK), 0.5 mM of each primer (Macrogen Inc., Seoul, South Korea), 1.25U/µl MangoTaq (Bioline, London, UK) and 2µl of DNA. The amplification series consisted of 35 cycles of the following: 95â—¦ C for 30 sec, 56â—¦ C for 30 sec and 72â—¦ C for 30 sec.”

Figure 6 and 7 – please, use a consistent order in both figures (i.e. control, naftifine and garlic extract or control, garlic extract and naftifine).

We changed Figure 7 (current Figure 4) according to Figure 6 (current Figure 3).

The naftifine was used as a reference compound, so its effect on colonies should not be discussed separately.

Figure 8 and 9 – parts A-D present components of control cells, while E and F present effects of garlic extract – please insert these information into pictures.

We inserted the information into pictures.

The abbreviation MIC is not explained.

MIC means Minimal Inhibitory Concentration. We defined the abbreviation at its first appearance in the text.

The abbreviations once introduced, should be consistently used in the text.

The abbreviations once introduced, were consistently used in the text.

In the results section, more details about antioxidant properties in in vitro studies should be provided.

Table 5 – it is completely chaotic: TAR, not TAC; abbreviations: INFL or INFLAM, DICLO or DICLOFENAC. In my opinion, INFL should be inserted just after CONTROL and then A. sativum and other groups.

We changed and corrected table 5 as suggested.

Why TAR has exactly the same value in all studied groups? Is it possible?

TAR had different values, but putting just 2 decimals it seemed to be identical. We changed that and put 4 decimals.

Units: equiv/L not mM or microM/L! MDA – in the methods, the unit nmol/ml of serum is used and in the Table there is nM/L!

We did the correction of the units.

What was measured as “total thiols”– only non-protein thiols or total thiols (protein and non-protein)? If total thiols (TSH) were determined, they included not only GSH but also protein thiols.

Total thiol levels were measured, and because glutathione (GSH) solutions were used to create a standard curve, serum SH concentration was stated in mmol GSH/mL.

Why was not statistical analysis performed on allicin group? Please, explain, why allicin group was studied.

Because allicin was the main compound from the extract, and it is was proved in other studies that it has antioxidant properties, we wanted to compare A. sativum effect with allicin. Statistical analysis was performed for allicin, but by mistake when we copied the table, we missed a row.

In the results section, the detailed values of p (i.e. p=1.22E-06) are not necessary, please give more general information (i.e. p<0.0001).

We did the suggested changes related to p value.

In the results on MDA, it is not true that A. sativum dilutions of 50% and 25% reduced MDA more than the undiluted extract 100% (100% extract – 5.54, while 25% - 5,79).

We did the correction of MDA results description.

Experimental design of the in vivo studies should be presented in a diagram. In the present form it is not clear and difficult to understand. There is no information about the dose of allicin.

Allicin dose was decided according to allicin concentration in 1ml of A.sativum extract 100%, respectively 450 μg/mL.

How were allicin and diclofenac administered to animals?

Allicin, diclofenac and A. sativum extracts were administrated by gavage.

What was a solvent for allicin and what for garlic extract?

Allicin was solved in water (450 μg/mL) and A. sativum extract was extracted in ethanol (1 g plant material in 1 mL of 20% ethanol).

What does it mean “The experiments were performed in triplicates” in reference to animals groups?

The in vivo experiments were repeated three times.

The antifungal effect of A. sativum extract – if 12% extract contains 120 mg /ml, why 5% contains 500 μg/ml. I think it is a mistake.

In the paragraph „ The antifungal effect of A. sativum extract”, the maximum inhibition for the garlic extract is compared to naftifine. At the concentration of 12%, the A. sativum extract contains 120 mg/mL, while the concentration of 5% refers to the concentration of naftifine hydrochloride (10mg/mL in Exoderil) used which contains 500 µg/mL.

Please, try to limit the number of references – 50 are too many in my opinion for experimental paper.

We reduced the references number from 50 to 36.

Language correction is suggested.

Reviewer 2 Report

English grammar and writing style are mostly fine; however, the manuscript would benefit from a read-through for punctuation (primarily comma usage) and article (a, an, the) usage.

Species names should be italicized, even when they occur in figure/table captions: lines 93, 95, 99, 106, 132, 138, 139, 150, 168, 171, 190, 193, 210, and 226.  When the species name appears in an italicized heading, it should not be italicized: lines 157, 200, and 338.

Commas should be replaced with decimals points (periods) in the following places: Line 169, naftifine concentration; Table 3, Colony a diameter for 8 and 10% garlic extract; Table 4, P b for 0.1% naftifine

Line 27: Please spell out the full genus name for Allium the first time that it appears in the abstract.

Line 78: The aim of a study should NEVER be to PROVE something.  Reword this.

Lines 87-88: Given the structural references, it would be useful to include pictures of the chemical structures of alliin and allicin.

Figure 3: How many times were these experiments run? What do the error bars represent (SD, SE, SEM, etc.)?

Lines 152-153: Although the information can be found in the Methods, the DNA sequences that were analyzed for the molecular confirmation of species identity should be specified when discussed in the Results and Discussion section.  Were the species suspected based on morphological identification the only ones with hits in the BLAST search?

Lines 237-260: This section is generally hard to read and somewhat confusing.  I suggest re-writing this section for clarity.  In addition, symbols (>, <, =) are missing in each occurrence of a given p-value.  Additional specific comments throughout this section:  Line 237 - It could be useful to provide a few clarifying details regarding the “acute inflammation model” as this is somewhat vague.  Line 241 - Specify the type of “animals” being used. Line 243 - Diclofenac should be briefly introduced and described, as it just appears out of nowhere.  Table 5:  What is INFLAM?  This is not mentioned elsewhere and seems like it is another drug/compound that is being tested.

Table 5:  Table would be easier to read if all of the comparisons for significance (a*, etc.) appeared on the line below the number measurement, rather than on the same line.  (For example, it would be easier to read if all comparisons appeared as they do in the TAC column, rather than how they appear in the OSI column.)  In addition, aligning the text in the treatment(?) column (CONTROL, A. sativum 100%, etc.) with the number values rather than being centered between the two lines of number values and significance comparisons would also make it easier to read.

Line 257: I believe Table 1 is incorrectly referenced here.

Line 284: “Effective” would probably be a better word here than “efficient”.

Line 399: Replace “proved” with “indicated.” You should never use “proved” in science, especially when referring to your own data.

Line 406: It is unnecessary to reference Figure 4A in the Methods.

Author Response

Tamara Milicevic

Assistant Editor

MDPI DOO

                                                                                                            October, 2019

Dear Mrs. Milicevic,

Subject: Submission of revised paper Molecules-620774

Thank you for the October 15th, 2019 e-mail enclosing the reviewer’s comments.

We would like to thank the reviewers for the careful and thorough reading of this manuscript and for the thoughtful comments and constructive suggestions. We have carefully reviewed the comments (shown below in italic) and have revised the manuscript accordingly. Our responses are given in a point-by-point manner below (dark blue text). Changes to the manuscript are shown in underline.

We look forward to hearing from you regarding our submission. We would be glad to respond to any further questions and comments that you may have.

Sincerely,

Response to Reviewer #2

English grammar and writing style are mostly fine; however, the manuscript would benefit from a read-through for punctuation (primarily comma usage) and article (a, an, the) usage.

We improved the writing style according to the suggestions.

Species names should be italicized, even when they occur in figure/table captions: lines 93, 95, 99, 106, 132, 138, 139, 150, 168, 171, 190, 193, 210, and 226. When the species name appears in an italicized heading, it should not be italicized: lines 157, 200, and 338.

The species names were italicized accordingly.

Commas should be replaced with decimals points (periods) in the following places: Line 169, naftifine concentration; Table 3, Colony a diameter for 8 and 10% garlic extract; Table 4, P b for 0.1% naftifine

We replaced the comas with decimal points as suggested.

Line 27: Please spell out the full genus name for Allium the first time that it appears in the abstract.

We spelt out the full name for Allium the first time that it appears in the abstract.

Line 78: The aim of a study should NEVER be to PROVE something. Reword this.

We reworded the phrase.

Lines 87-88: Given the structural references, it would be useful to include pictures of the chemical structures of alliin and allicin.

We agree, we added the chemical structures of the two compounds in Figure S1, besides the MS spectra. The section Phytochemical analysis of the garlic extract was transferred to the Supplementary material.

Figure 3: How many times were these experiments run? What do the error bars represent (SD, SE, SEM, etc.)?

The experiments were run three times (n=3) independetly, the erros represent SD and it was indicated in the text (current Figure S3).

Lines 152-153: Although the information can be found in the Methods, the DNA sequences that were analyzed for the molecular confirmation of species identity should be specified when discussed in the Results and Discussion section. Were the species suspected based on morphological identification the only ones with hits in the BLAST search?

The molecular confimation is discussed in the Results and Discussion section. We added the following text in order to give more details regarding the BLAST search.

Line 158: „The Percentage identity and the other alignment indices (Max score, Total score, and Query coverage) were the highest (100%) when the DNA sequences were compared to M. guilliermondii and R. mucilaginosa sequences present in the NCBI database, confirming that the isolated species have the most similar DNA sequences to the above-mentioned species.”

Lines 237-260: This section is generally hard to read and somewhat confusing. I suggest re-writing this section for clarity. In addition, symbols (>, <, =) are missing in each occurrence of a given p-value.

We rewrote more clearly this section and put symbols for p significance.

Additional specific comments throughout this section: Line 237 - It could be useful to provide a few clarifying details regarding the “acute inflammation model” as this is somewhat vague.

In the methods chapter we described the acute inflammation model induced by turpentine oil.

Line 241 - Specify the type of “animals” being used.

In the methods chapter we mentioned that the experiments were carried out on male Albino Wistar rats.

Line 243 - Diclofenac should be briefly introduced and described, as it just appears out of nowhere.

We added: „The effects were compared to diclofenac, an anti-inflammatory drug, and to allicin, the main compound from the phytochemical analysis of the tested extract”.

Table 5: What is INFLAM? This is not mentioned elsewhere and seems like it is another drug/compound that is being tested.

INFLAM abbreviation for the inflammation group was explained when first mentioned in the text. We added Figure 8 (Flowchart) to better explain the in vivo experiments.

Table 5: Table would be easier to read if all of the comparisons for significance (a*, etc.) appeared on the line below the number measurement, rather than on the same line. (For example, it would be easier to read if all comparisons appeared as they do in the TAC column, rather than how they appear in the OSI column.) In addition, aligning the text in the treatment(?) column (CONTROL, A. sativum 100%, etc.) with the number values rather than being centered between the two lines of number values and significance comparisons would also make it easier to read.

We did the suggested changes by putting the significance on the line below and centered.

Line 257: I believe Table 1 is incorrectly referenced here.

We corrected the referenced table from Table 1 to Table 5 (current Table 3).

Line 284: “Effective” would probably be a better word here than “efficient”.

We replaced the word as suggested.

Line 399: Replace “proved” with “indicated.”You should never use “proved” in science, especially when referring to your own data.

We replaced the word as suggested.

Line 406: It is unnecessary to reference Figure 4A in the Methods.

We deleted the reference to Figure 4A in the Methods chapter.

Round 2

Reviewer 1 Report

I have read the revised version of the manuscript: “Allium sativum extract chemical composition, antioxidant activity and antifungal effect against Meyerozyma guilliermondii and Rhodotorula mucilaginosa causing onychomycosis”. The authors have corrected the manuscript and have addressed the suggestions. I am rather pleased with the improvements, however I think the papers still needs minor corrections.

Please insert paragraph numbering according to the rules of the journal (1. Introduction, 2. Results, etc) Line 128 and line 170 – in my opinion “A. sativum” should be in italic. Figure 3 legend – line 140 double bracket, I suggest put (D) after (without colony) Figure 4 legend – line 140, I suggest put (H) after (without colony) Table 1 legend -line 144 „=” should be deleted Table 1 legend – line 165, the word “Legend” is not necessary (it is absent in Table 3 legend) Table 1 legend -line 144 the concentration of naftifine solution given in bracket is not necessary here (it is given in another place, moreover various concentrations of naftifine were used) Table 3 – I understand that GSH was used to prepare standard curve, however the unit “TSH mmol/ml” should be used instead GSH mmol/ml. Line 228 - „=” should be deleted Material and Methods section – the first paragraph (titled i.e. Chemicals) should contain information about used chemicals (i.e. naftifine, diclofenac, allicin was synthetized as is described in supplementary material, etc) Material and Methods section – the first two current paragraphs should be one titled “Allium sativum extract” The paragraph “Phytochemical analysis” is present now in supplementary material, so in the main text it should be only mentioned about it. Line 382 – the title “ In vitro antioxidant effect” will be better (without analysis) Line 387 – the title “ In vivo antioxidant effect” will be better (without evaluation) Line 388 – the chapter “experimental design should be omitted

Author Response

Milena Amidzic

Assistant Editor

MDPI DOO

                                                                                                            October, 2019

Dear Mrs. Amidzic,

Subject: Submission of revised paper Molecules-620774

Thank you for the October 25th, 2019 e-mail enclosing the reviewer’s comments.

We would like to thank the reviewers for the careful and thorough reading of this manuscript and for the thoughtful comments and constructive suggestions. We have carefully reviewed the comments (shown below in italic) and have revised the manuscript accordingly. Our responses are given in a point-by-point manner below (dark blue text). Changes to the manuscript are shown in underline.

We look forward to hearing from you regarding our submission. We would be glad to respond to any further questions and comments that you may have.

Sincerely,

Response to Reviewer #1

I have read the revised version of the manuscript: “Allium sativum extract chemical composition, antioxidant activity and antifungal effect against Meyerozyma guilliermondii and Rhodotorula mucilaginosa causing onychomycosis”. The authors have corrected the manuscript and have addressed the suggestions. I am rather pleased with the improvements, however I think the papers still needs minor corrections.

Please insert paragraph numbering according to the rules of the journal (1. Introduction, 2. Results, etc).

We inserted paragraph numbering according to the rules of the journal.

Line 128 and line 170 – in my opinion “A. sativum” should be in italic.

Figure 3 legend – line 140 double bracket, I suggest put (D) after (without colony). Figure 4 legend – line 140, I suggest put (H) after (without colony)

We made the suggested changes.

Table 1 legend -line 144 „=” should be deleted Table 1 legend – line 165, the word “Legend” is not necessary (it is absent in Table 3 legend) Table 1 legend -line 144 the concentration of naftifine solution given in bracket is not necessary here (it is given in another place, moreover various concentrations of naftifine were used).

We made the suggested changes.

Table 3 – I understand that GSH was used to prepare standard curve, however the unit “TSH mmol/ml” should be used instead GSH mmol/ml.

Line 228 - „=” should be deleted

We made the suggested changes.

Material and Methods section – the first paragraph (titled i.e. Chemicals) should contain information about used chemicals (i.e. naftifine, diclofenac, allicin was synthetized as is described in supplementary material, etc)

Material and Methods section – the first two current paragraphs should be one titled “Allium sativum extract”.

We made the suggested change.

The paragraph “Phytochemical analysis” is present now in supplementary material, so in the main text it should be only mentioned about it.

Line 382 – the title “ In vitro antioxidant effect” will be better (without analysis).

Line 387 – the title “ In vivo antioxidant effect” will be better (without evaluation).

Line 388 – the chapter “experimental design should be omitted.

We made the suggested changes.